# A model of naturalistic decision making in preference tests

**John Ksander**[1,2], **Donald B. Katz**[1,2], **Paul Miller**[1,3]*

**1** Volen National Center for Complex Systems, Brandeis University, Waltham, Massachusetts, United States of America, **2** Department of Psychology, Brandeis University, Waltham, Massachusetts, United States of America, **3** Department of Biology, Brandeis University, Waltham, Massachusetts, United States of America

* pmiller@brandeis.edu

**Data Availability Statement:** Data are available at https://github.com/johnksander/naturalistic-decision-making.

**Funding:** PM and DBK received funding and salary coverage from NIH and the BRAIN Initiative via R01 NS104818. JK received salary coverage from

## Abstract

Decisions as to whether to continue with an ongoing activity or to switch to an alternative are a constant in an animal's natural world, and in particular underlie foraging behavior and performance in food preference tests. Stimuli experienced by the animal both impact the choice and are themselves impacted by the choice, in a dynamic back and forth. Here, we present model neural circuits, based on spiking neurons, in which the choice to switch away from ongoing behavior instantiates this back and forth, arising as a state transition in neural activity. We analyze two classes of circuit, which differ in whether state transitions result from a loss of hedonic input from the stimulus (an "entice to stay" model) or from aversive stimulus-input (a "repel to leave" model). In both classes of model, we find that the mean time spent sampling a stimulus decreases with increasing value of the alternative stimulus, a fact that we linked to the inclusion of depressing synapses in our model. The competitive interaction is much greater in "entice to stay" model networks, which has qualitative features of the marginal value theorem, and thereby provides a framework for optimal foraging behavior. We offer suggestions as to how our models could be discriminatively tested through the analysis of electrophysiological and behavioral data.

## Author summary

Many decisions are of the ilk of whether to continue sampling a stimulus or to switch to an alternative, a key feature of foraging behavior. We produce two classes of model for such stay-switch decisions, which differ in how decisions to switch stimuli can arise. In an "entice-to-stay" model, a reduction in the necessary positive stimulus input causes switching decisions. In a "repel-to-leave" model, a rise in aversive stimulus input produces a switch decision. We find that in tasks where the sampling of one stimulus follows another, adaptive biological processes arising from a highly hedonic stimulus can reduce the time spent at the following stimulus, by up to ten-fold in the "entice-to-stay" models. Along with potentially observable behavioral differences that could distinguish the classes of networks, we also found signatures in neural activity, such as oscillation of neural firing rates and a rapid change in rates preceding the time of choice to leave a stimulus. In summary,

the Swartz Foundation #2020-13. Computational resources were provided by the Brandeis HPCC which is partially supported by the Brandeis Center for Bioinspired Soft Materials, and NSF MRSEC, DMR-2011846. The funders had no role in study design, data collection and analysis, decision to publish, or preparation of the manuscript.

**Competing interests:** The authors have declared that no competing interests exist.

our model findings lead to testable predictions and suggest a neural circuit-based framework for explaining foraging choices.

## Introduction

Decisions as to whether to stay in a current situation or switch to a different one face all animals continuously. Such "should I stay, or should I go now" decision-making has been studied most in the context of foraging [1–4], in which an animal decides whether to continue to stay at a current source of food or to seek an alternative source. In foraging studies, the current source of food necessarily yields diminishing returns, such that at some point in time it is optimal for the animal to seek a higher-quality option. Furthermore, this tendency appears "baked in" to animal behavior: while food sources in laboratory food preference tests remain in constant supply, animals nonetheless typically switch back and forth between two or more alternatives.

Here, we simulate a model of such switching back and forth in terms of transitions between quasi-stable attractor states of neural activity. Quasi-stable attractor states [5] are patterns of neural activity that are essentially self-sustaining but limited by adaptation processes or fluctuations that eventually lead to a loss of stability and a transition to a new pattern of activity [6]. Evidence for such attractor states is most abundant in neural activity arising from perception [7,8], with quasi-stability most apparent in the switching between bistable percepts. More recently, experimental evidence for transitions between attractor states has been found [9,10] and modeled [11–13] in perceptual decision-making tasks. For those tasks, the attractor states have been proposed to represent one of two possible percepts for a given stimulus [14], or the absence of a decision (i.e., an undecided state)[11,13]. In the more naturalistic decision-making model we present here, two activity states represent the ongoing choice to either stay with a current stimulus or switch to a new stimulus [4]. It is revealing that, according to behavioral data [15], the distribution of bout durations, i. e., when an animal stays at a stimulus (corresponding to the durations of the "stay" state in our model) is approximately exponential, which is a hallmark of noise-induced transitions between discrete states [16].

Our simulations differ from those aimed to model the circuitry underlying perceptual decision making in a second way. In preference tests the stimuli are separated in time, rather than via distinct neural pathways. For example, two tastes being compared could simply be different concentrations of sodium chloride concentration, which yields an inverted-U palatability response [17]. The two stimuli excite identical neurons during successive sampling bouts, with neurons indicating high palatability during one bout being the same as those indicating high palatability during the next bout. In this manner, a preference task is more akin to decision making tasks requiring a sequential comparison of a single parametric quantity [18–20], although in this paper the quantity being compared is palatability and the decision is an ongoing one of how long to stay at the stimulus.

Our work compares performance between two model classes, distinguished by their parameter settings. In one class, the system is inherently "fickle" in that the "stay" state is unstable; an animal operating by this model would stay for only a very short duration of time unless neural activity is stabilized by input indicative of a highly positive hedonic stimulus. One could say in these networks that a delicious stimulus entices the animal to stay, but otherwise the animal leaves. In the other class of model, the system is "committed" in the sense that the "stay" state is highly stable, such that an animal operating by this model would stay for relatively long durations, until input suggesting stimulus aversiveness causes a transition away from that

state. One could say in these networks that the stimulus causes the animal to leave, but that otherwise the animal stays.

We produce multiple versions of each class of model, in order to assess how they could be reliably distinguished in behavioral or electrophysiological data [4,21,22]. Our simulations suggest that any observation of strongly reduced sampling-bout durations at one stimulus following the sampling of a highly hedonic stimulus would indicate animals operating with the "entice to stay" class of network. Also, in the "entice to stay" class of network, neural activity appears to change gradually over more than 100ms preceding a choice to leave a stimulus, whereas in the "repel to leave" network, the activity change is more sudden, but preceded by oscillations in the beta range. Therefore, we suggest that choice-aligned averaging of neural spike trains, combined with appropriate analysis of animal behavior in preference tasks [23] can distinguish the two classes of models.

## Results

### Network characteristics

Our initial goal was to set up a network with two distinct activity states, one representing an animal continuing to sample a stimulus, or "staying", the other representing the choice to "switch" elsewhere. Fig 1 shows the behavior of such a network in the absence of stimulus (i.e. when the network receives only noisy background input). The raster plot in Fig 1B illustrates how the network's activity abruptly shifts back-and-forth between two states of activity, which are quasi-stable attractor states. When the excitatory neurons in the "stay" population become highly active, those in the "switch" population become silent, and vice versa. Fig 1C shows these same data in terms of each population's averaging spike rates. Noisy fluctuations in the spiking activity drive these transitions between stay and switch states. While Fig 1 indicates the quasi-stable nature of the circuit in the absence of any stimulus, when we include a stimulus in the circuit simulation, activation of the "switch" population represents the choice to end a bout of sampling.

Our model incorporates synaptic depression, whereby a rapid series of neural spikes depletes the supply of synaptic vesicles and therefore reduces synaptic efficacy. A docking process, which renders available vesicles release-ready is relatively fast (a few hundred ms), so we denote the fraction of docked vesicles as $D_{fast}$. Recycling and filling of vesicles to regenerate the activity-depleted pool is slower (many seconds), however, so we denote the fraction available for docking as $D_{slow}$. Fig 1D–1E show how the population-averages of these depression variables change as the model switches between activity states shown in Fig 1B–1C. Specifically, Fig 1D illustrates how docking sites release their vesicles during sustained spiking, and then quickly become replenished from the reserve pool after activity ceases. Fig 1E illustrates how activity empties reserve vesicle pools, as the vesicles stored there replenish empty docking sites. The reserve vesicle pools then slowly recoup their losses after the cells stop firing and vesicle regeneration outpaces the losses to docking sites. Note that reserve vesicle pools sometimes fail to refill fully before the next state transition. A long active state will deplete the reserve vesicle pool to a degree that those reserves cannot fully recover from without a lengthy inactive period.

### Parameter sweep and example networks

In order to ensure that any results of our study are robust to parameter variations, we generated a number of distinct "entice to stay" and "repel to leave" networks before assessing qualitative differences between the two classes' behavior. To this end, we first measured the mean

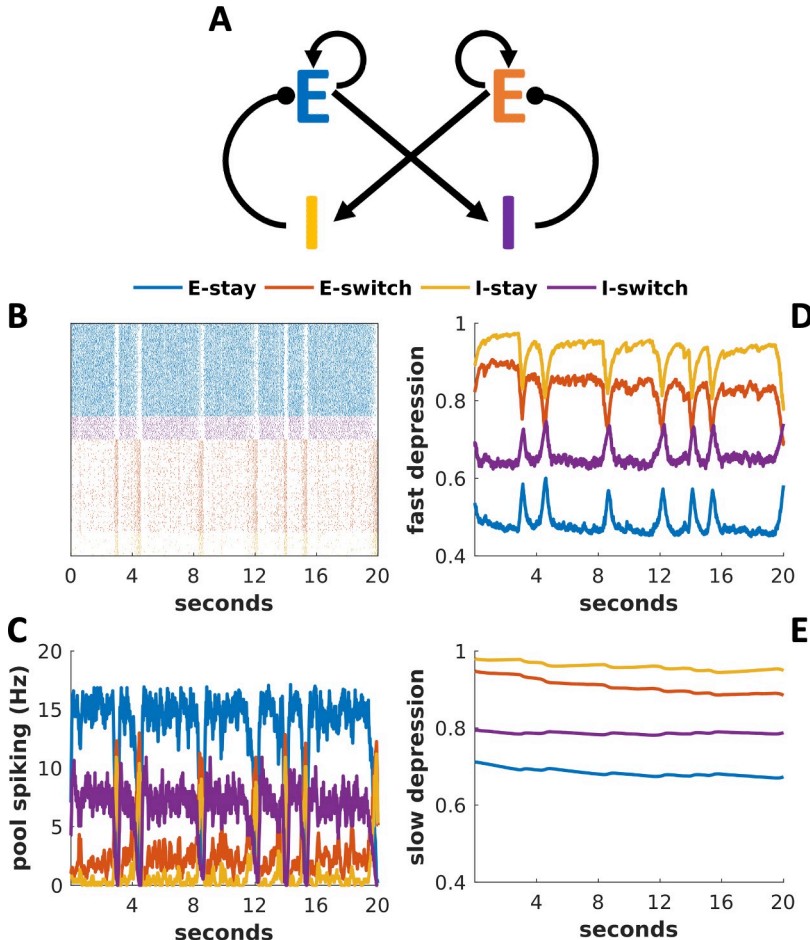

**Fig 1. Network in a default mode without stimulus possesses two quasi-stable attractor states.** Example data from a fast ("entice to stay") network in the absence of stimulus. (A) Circuit diagram, indicating cross-inhibition between excitatory populations of cells labeled E, inhibited by their corresponding inhibitory population labeled I. (B) Spike rasters from model neurons arranged in rows grouped by their corresponding populations and color-coded as in (A). Each dot represents one spike from the model neuron. (C) Mean firing rate of each population as a function of time. (D) Dynamics of the fast-depression variable. (E) Dynamics of the slow-depression variable. (C-E) Line color indicates which population's average variable is plotted with the color code as in (A).

durations of "stay" states in multiple networks generated across a five-fold range of excitatory-to-inhibitory (E-to-I) and inhibitory-to-excitatory (I-to-E) connection strengths.

Fig 2 indicates the range of parameters for which we found state-switching behavior, with mean duration of the "stay" state indicated in color in Fig 2. We chose five networks of each of the two classes, with "entice to stay" networks having the shortest intrinsic "stay" state durations of under two seconds (solid symbols in Fig 2) and "repel to leave" networks having the longest intrinsic "stay" state durations of over 100 seconds (open symbols in Fig 2). We typically paired networks (same symbol shape in Fig 2) with either the same E-to-I connection strength or the same I-to-E connection strength. Across the range of parameters, firing rates of excitatory and inhibitory neurons were at levels compatible with those of cortical neurons in an active state (near 10 Hz for excitatory cells and up to 60 Hz for inhibitory cells, S1 Fig). These results demonstrate that our circuit produces quasi-bistable state-switching behavior with realistic spike-rates and that this behavior is robust to variation in synaptic connection strengths (see also S2 Fig).

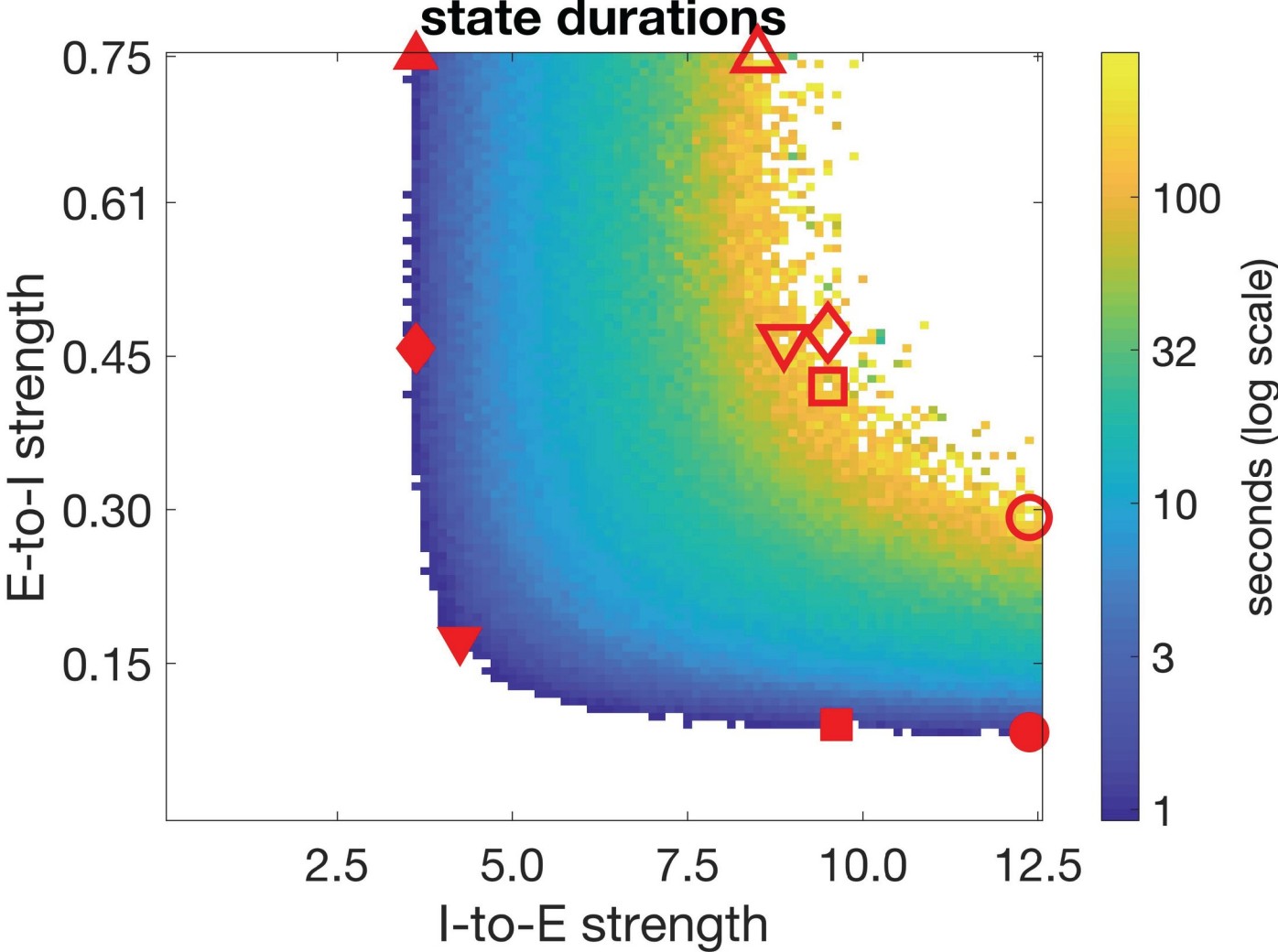

**Fig 2. Intrinsic stability of the "stay" state varies with connection strengths.** Average duration of the "stay" state in distinct networks with different synaptic connection strengths, without stimulus. Symbols indicate parameters of the five example networks, with solid symbols representing fast-switching "entice-to-stay" networks and open symbols representing slow-switching "repel to leave" networks. Identical shapes indicate the pairs used for comparison.

We next performed simulations with all ten example networks (symbols in Fig 2). Our results generalized well across these parameterizations, so for simplicity we first describe data arising from one pair of networks from each class–specifically, from the networks with square markers in Fig 2. In the supplementary materials (S3 and S4 Figs) we detail the simulation results from all ten networks in cases where they are not all included in the main text, for completeness.

Since, by definition, sampling bouts are in the presence of a stimulus, and their durations are a key behavioral measurement, our first goal was to find "baseline" stimulus values that reproduced similar mean durations of the "stay state" for each network. We therefore defined a neutral stimulus for each network, as one which produces mean "stay state" durations between eight and nine seconds, in the middle of the observed ranges of typical bout durations for a rat licking taste stimuli from a spout [15,24].

Fig 3 illustrates how application of the neutral stimulus to intrinsically fast-switching networks via excitation of neurons driving the "stay" state (upper left) realizes the "entice to stay"

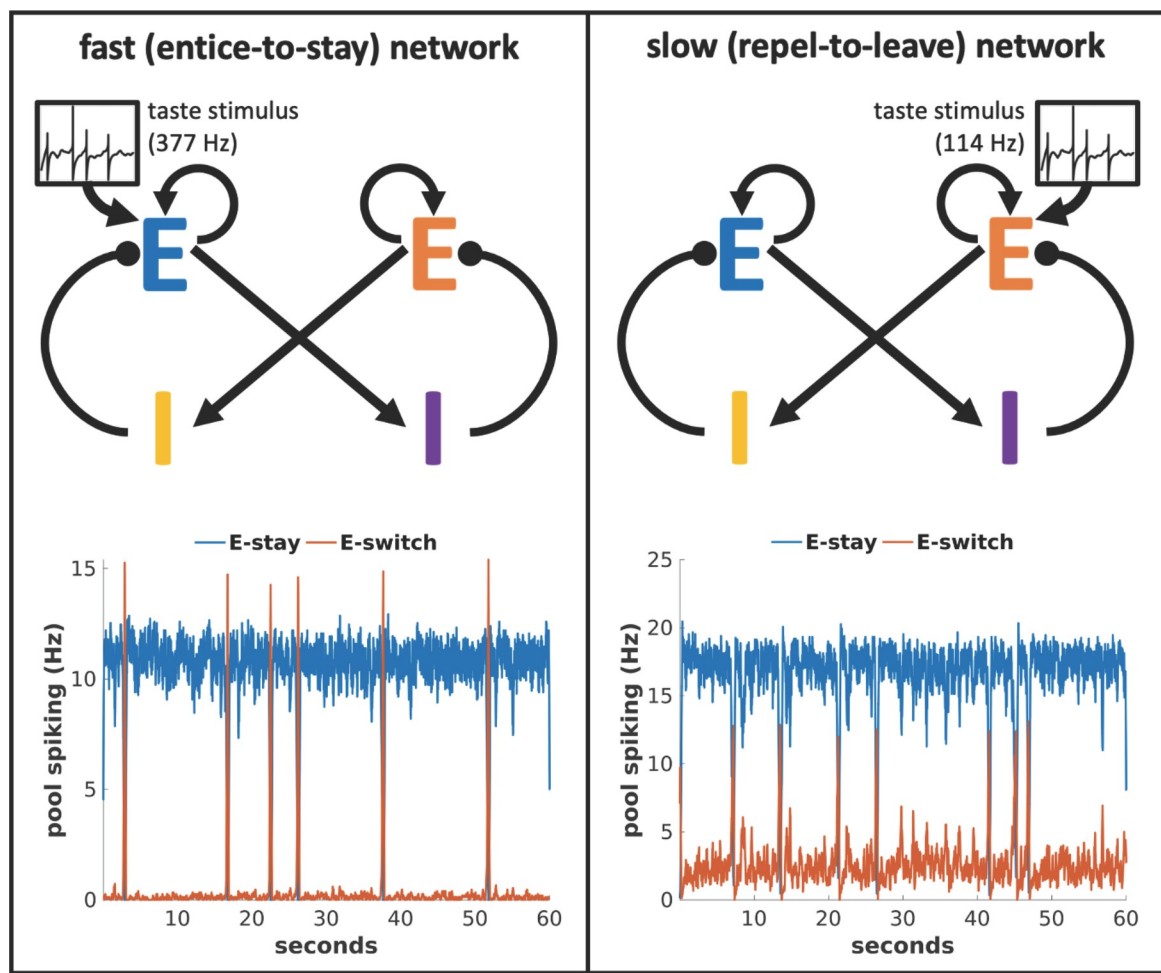

**Fig 3. In response to a taste stimulus, neural activity in the two classes of network can lead to equal mean sampling durations.** Left: Representing a neutral taste stimulus via excitatory input to "stay"-promoting neurons in a fast (entice-to-stay) network slows state transitions. Right: Whereas representing a neutral taste stimulus via excitatory input to "switch"-promoting neurons in a slow (repel-to-leave) network quickens state transitions to produce the same switching dynamics.

operation of a stimulus. The stimulus slows the network's transitions from stay to leave from its stimulus-free duration of under two seconds such that the network achieves the desired mean bout duration (lower left). Conversely, application of the neutral stimulus to intrinsically slow-switching networks via excitation of neurons driving the "switch" state (upper right) realizes the "repel to leave" operation of a stimulus. In these networks, the stimulus quickens the network transition such that the network's stimulus-free mean durations in the "stay" state of a few minutes are shortened to the desired mean bout duration (lower right). In these distinct manners, the same mean behavior to a given, neutral, external stimulus can arise from either class of network.

We do not simulate the behavior of the animal between bouts of tasting, but once the activity state indicates a switch of stimuli, we remove all stimulus input in the model to indicate movement away from the food source. We subsequently reduce input to the excitatory "switch"-producing cells to indicate completion of the movement and ensure commencement of the next sampling bout with the "stay" state. In the following sections the stimulus during the subsequent bout corresponds to the alternative tastant, which in many cases has a different palatability so produces a spike train of a different rate, as indicated in Fig 4.

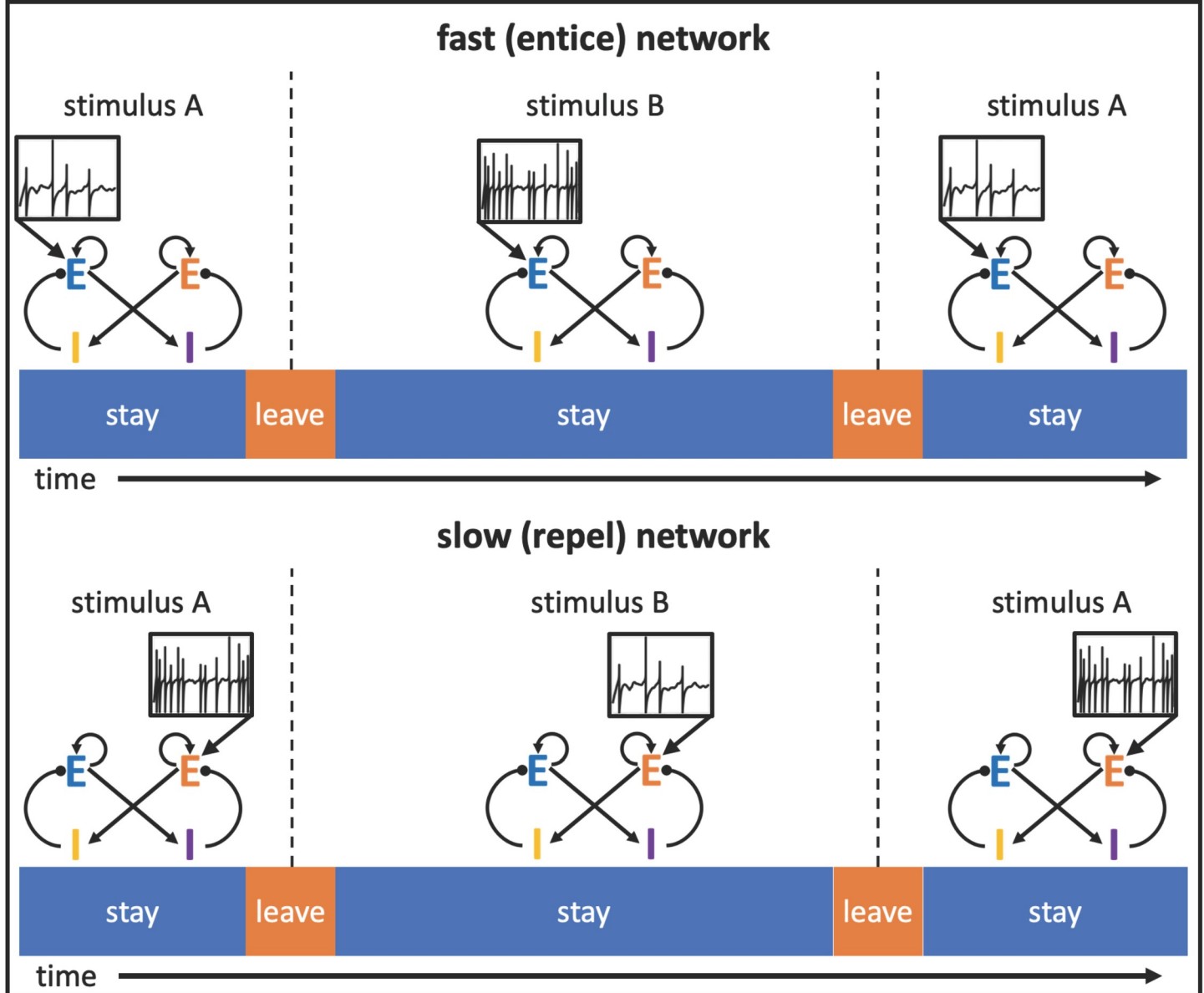

**Fig 4. Example of a taste preference task simulation.** Animals alternate between a less palatable stimulus A and a more palatable stimulus B. In the intrinsically fast-switching "entice-to-stay" network, the increased palatability arises from greater excitatory input to the excitatory "stay" pool of neurons, whereas in the intrinsically slow-switching "repel-to-leave" network, the increase in palatability is produced by a reduction of excitatory input to the excitatory "switch" pool of neurons.

### Taste preference behavior

In taste preference tests, consuming one stimulus comes at the expense of another, primarily because the animal has limited desire and time for consuming food or liquid. Therefore, as in any time-limited task, there is an inherent competition in that more time spent doing A means less time spent doing B. However, a key unanswered question is whether the nature of one taste stimulus impacts the neural and behavioral responses to a subsequent, second taste stimulus. Association with a more palatable stimulus via temporal proximity could, in principle, enhance the perceived palatability of an alternative stimulus. However, the qualitative results of foraging studies as encoded in the Marginal Value Theorem, suggest the reverse is true.

That is, during foraging, animals spend less time at a food source if there are better sources easily reached elsewhere. So, in a food preference task it is possible that higher stimulus palatability of one food source leads to a shortening of bout durations at the alternative food source. Therefore, we assessed whether such bout-to-bout competition between stimuli would arise in our networks.

Fig 4 illustrates how we simulate the taste preference task, such that the model switches between stimuli of different palatability (see *Taste preference task simulations* in the *Methods* section for full details). A key distinction between the two classes of models is apparent even before examining their performance, in the different representations of stimulus palatability. When testing the intrinsically fast-switching "entice-to-stay" networks, we simulate a more palatable tastant by increasing the excitatory input to excitatory neurons representing the "stay" activity state; this increases "stay" state durations. When testing the intrinsically slow-switching "repel-to-leave" networks, meanwhile, we simulate that same tastant by reducing the excitatory input to excitatory neurons representing the "switch" activity state; this also increases "stay" state durations. We identify the time spent in the "stay" state as the duration of active bouts of food sampling, and therefore proportional to the amount of food consumed (the basic behavioral measure of palatability). In these distinct manners, both classes of network represent taste stimuli of varying palatability via varying afferent firing rates.

To assess the impact of one stimulus on the other, we measured the mean bout durations (our measure of stimulus palatability) of stimulus A as a function of the mean bout duration of stimulus B. For these simulations, we fixed one of a pair of stimuli (stimulus A, baseline) and varied the palatability of the second stimulus (stimulus B, varied).

The results of these analyses, which are shown in Fig 5, differentiate network type. For both classes of network, competition between bout durations arises—with increased palatability and bout duration of the varied stimulus B, the bout durations of the fixed stimulus A decrease. For the intrinsically slow switching "repel-to-leave" networks (red points in Fig 5), the competition can produce a factor of two reduction in the sampling bout durations at the unchanged stimulus A. However, the competition has a much larger impact on the intrinsically fast switching "entice-to-stay" networks (blue points in Fig 5), such that sampling bout durations at the fixed stimulus A can decrease by up to ten-fold over the same range of B variation.

To explain this more strongly competitive interaction between successive stimuli in the "entice-to-stay" network (Fig 5, lower panels, solid curves), it is worth considering differences in how the relative inputs arising from hedonic stimuli impact these networks. In the "entice-to-stay" network, a positively hedonic stimulus B is simulated in terms of large amounts of input to the "stay" neurons in the network. These neurons receive strong input and fire at high rates as they represent an animal sampling the positively hedonic stimulus for long durations. A long duration of relatively high firing rates, meanwhile, is exactly the network activity that is expected to deplete the reserve vesicle-pool and maximize adaptation at the circuit-level. Note that the converse does not apply–that with a highly aversive stimulus B, and lower firing rates of the "stay" neurons, the sampling durations in response to stimulus A do not increase significantly.

In the "repel-to-leave" network, meanwhile, the coincidence of extended duration higher than baseline firing rate never arises. The increased hedonic value of B is instead achieved *via* reduced input to the "leave" neurons in the network, such that long sampling of a positively hedonic stimulus causes a reduction of synaptic depression in the circuit. One might expect that synaptic depression therefore induces competition in the opposite case—when stimulus B is highly unpalatable, the input to the "leave" neurons, and therefore their firing rates, are high —but in such a scenario the bout durations are too short for the reserve vesicle-pool to be depleted significantly.

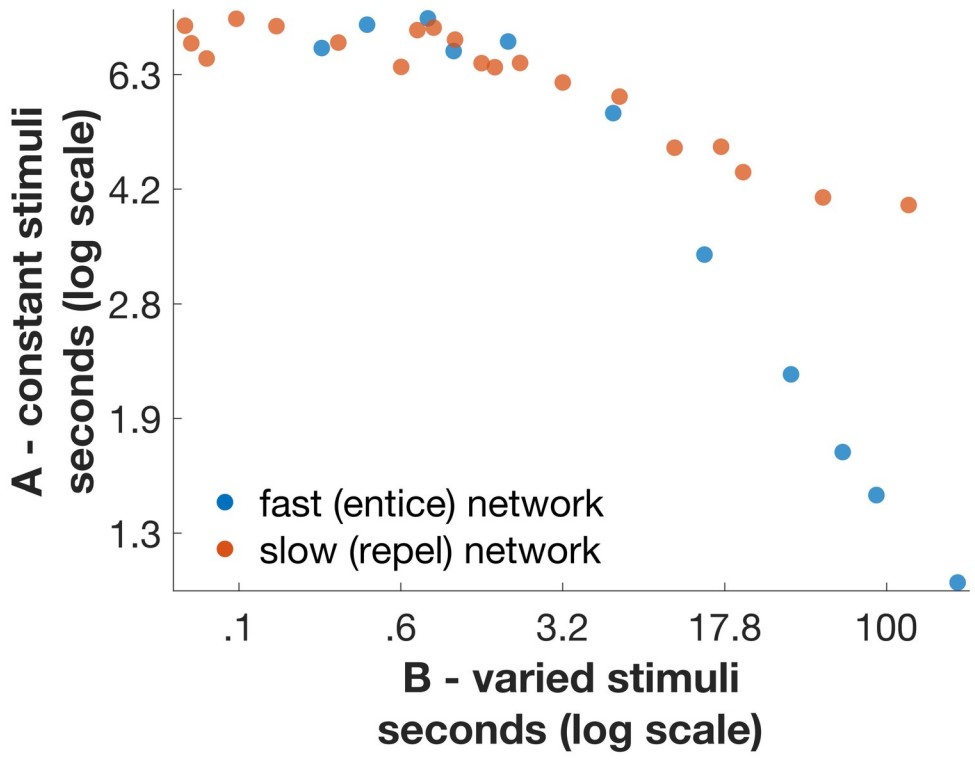

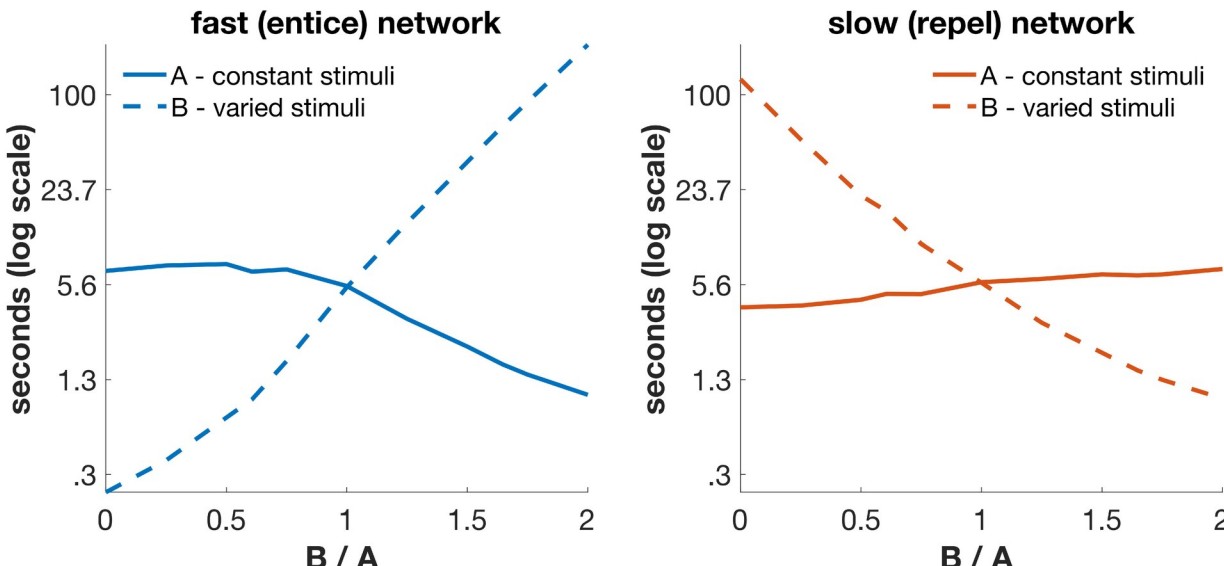

**Fig 5. Mean duration of sampling bouts in simulated taste preference tests depends on the alternative stimulus in a competitive manner.** (Top) Behavior of intrinsically fast-switching ("entice-to-stay") networks is shown as blue points, and intrinsically slow-switching "repel-to-leave" networks as red points. In both networks the greater the palatability of stimulus B, the lower the perceived palatability of the unchanging stimulus A, but the bout-to-bout competition is much stronger in the "entice-to-stay" network when stimulus B is highly palatable. (Bottom) Mean durations of the "stay" state are plotted as a function of the ratio of the inputs corresponding to the two stimuli, A and B. Each pair of a values for a given ratio provides one data point in the top panel. The input for stimulus A is held fixed while the input for stimulus B is varied as a parameter to produce distinct values along the x-axis. The two classes of network are distinguished in whether the input excites the "stay" pool (entice-to-stay network, lower left) or the "leave" pool (repel-to-leave network, lower right) and whether, respectively, high input or low input for stimulus B reflects the most hedonic stimulus B. Competition is revealed as the duration of responses to stimulus A (solid line) changes in the absence of any change in inputs for stimulus A (most evident in the left panel).

In conclusion, Fig 5 demonstrates differences between the behavior of the intrinsically fast-switching "entice-to-stay" network and that of the intrinsically slow-switching "repel-to-leave" network, with strong competition between bout durations only possible in the "entice-to-stay" class of networks. Moreover, our modeling reveals an asymmetry in the nature of competition, whereby a highly hedonic alternative stimulus reduces the duration of sampling bouts, but a highly aversive alternative stimulus produces bout durations that are little different from those with a neutral alternative stimulus, or no alternative at all.

## Network transition dynamics

One of our modeling goals was to assess whether these two classes of network show systematic differences in neural activity, in addition to the behaviorally observable quantitative differences in the degrees of competition shown in Fig 5. One structural difference has to do with activity in the circuit causes a transition away from stimulus sampling. In the case of the intrinsically fast-switching "entice-to-stay" networks, this transition is caused by a dip of input to the stay population, whereas in intrinsically slow-switching "repel-to-leave" networks the cause is a rise of input to the leave population. Given that neurons in any circuit are strongly interconnected in dynamic systems such as those described here, it is not clear that any change in one set of neurons could be measured independently of changes in other sets of neurons. However, we hypothesized that by aligning neural activity to the state-transition points we might make progress in uncovering distinctive signatures in the neural dynamics of our models that would allow us to distinguish the two classes of network using electrophysiological data.

Fig 6 presents such transition-aligned activity in examples of intrinsically fast-switching (entice-to-stay) networks (left) and intrinsically slow-switching (repel-to-leave) networks (right). While many differences between these examples (e.g. net firing rates), proved inconsistent across broader swaths of the two classes of models, two distinguishing features of the pre-transition activity are reliable: First, whereas in "entice-to-stay" networks (left), activity in the excitatory cells that cause the transition (E-switch, red) increases gradually during the 200ms before leave decisions, in "repel-to-leave" networks (right) E-switch activity remains low until only a few tens of milliseconds before the decision (at which point there is a sudden and rapid increase); second, the neurons inhibiting the transition (I-switch cells, purple) begin oscillating in the pre-transition period for "repel-to-leave" networks, but the same does not occur in "entice-to-stay" networks. These observations suggest measured neural activity (such as that acquired from gustatory cortex) could enable us to determine which class of network is active in a rodent's brain, if aligned to the time-point ending a bout of sampling.

## Discussion

It is worth comparing and contrasting the model we present here of stay-versus-switch decision making, based on a naturalistic task, with other models of decision making in systems neuroscience [11,13,14,25–31]. In particular, over the past few decades, much investment has been spent in studying winner-takes-all models of decision making that underlie the behavior of two-alternative forced choice tasks [25,29,32], usually in the context of perceptual decisions [33,34]. The framework of such behavioral tasks is trial-based, where the experimenter chooses the stimuli, and the subject makes a single, final choice in each trial. The relative preference for one alternative can be obtained after accumulating many discrete trials. Conversely, in naturalistic tasks such as food preference tests or foraging [1,35], the subject selects the stimulus, but the selection is not final. The choice is inextricably linked to the stimulus and is ongoing and dynamic.

Moreover, models of the two types of decision-making tasks must be distinct in two ways. First, any competition between stimuli must occur across time in a naturalistic task, since only

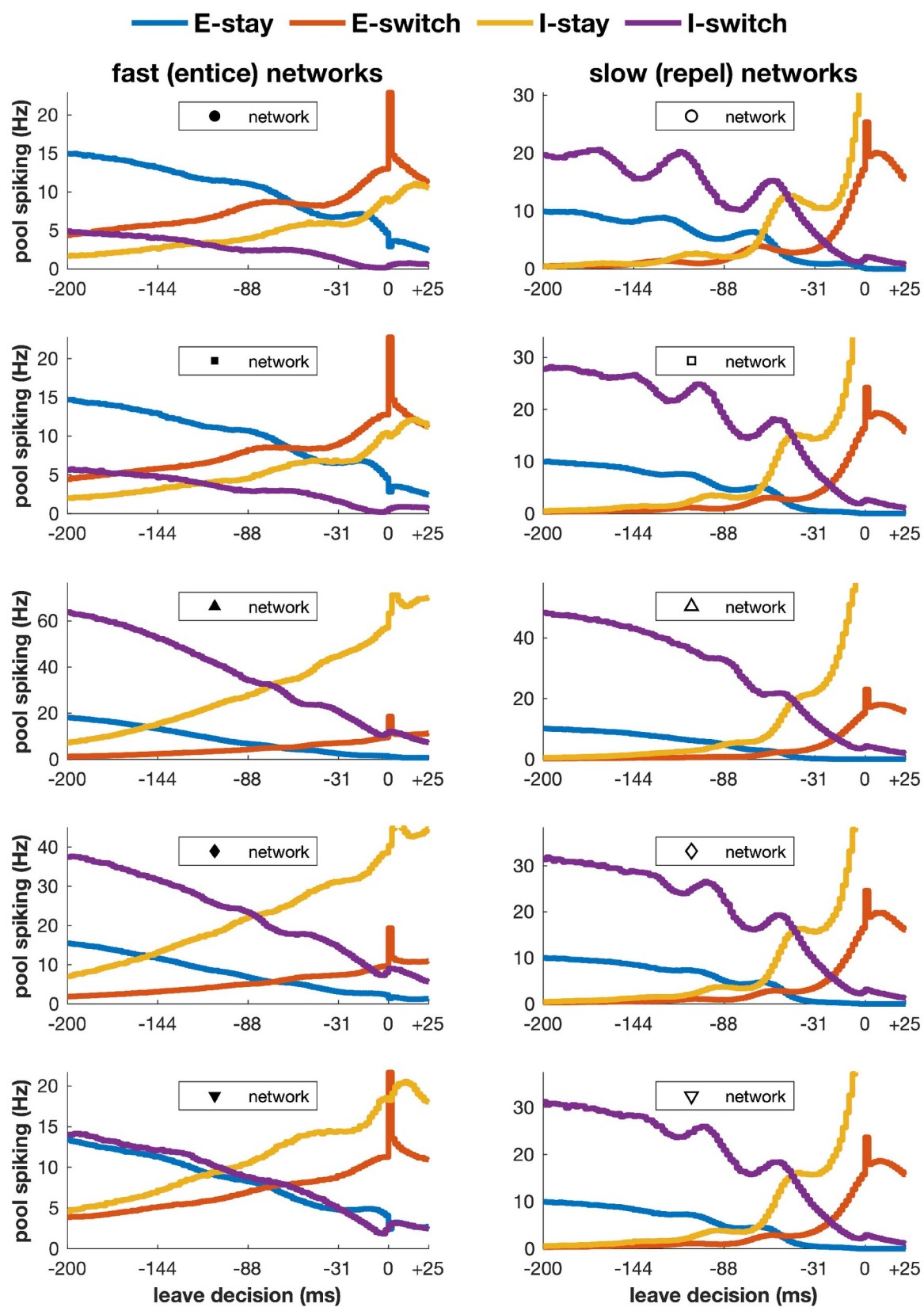

**Fig 6. Neural activity preceding transitions to switch away to a new stimulus.** A transition is detected when synaptic output of the E-switch neurons significantly exceeds that of the E-stay neurons, so the transition time (0 ms) is inevitably marked by a spike in activity of the E-switch neurons. A decline in activity of neurons that promote the "stay" (E-stay, blue) or inhibit the switch (I-switch, purple) combined with a rise in activity of neurons that promote the switch (E-switch, red) or inhibit the stay (I-stay, yellow) precedes the detected transition. Left: the inherently fast-switching, "entice-to-stay" networks. Right: the inherently slow-switching "repel-to-leave" networks.

one stimulus is present at any given time. Second, and importantly, the property of each of the two stimuli being evaluated is likely to provide input to the decision-making circuitry via the same neural pathway. Whereas in the most common perceptual decision-making paradigms, and therefore in models thereof, the two competing inputs correspond to opposing stimuli (such as directions of motion) and so excite different groups of neurons. In such perceptual tasks, the competition between stimuli can arise via a direct inhibitory interaction between the simultaneously active distinct groups of neurons. By contrast, in food preference tests the hedonic, appetitive neural response to one stimulus must compete with a similar hedonic, appetitive neural response to a temporally separated stimulus. Therefore, these naturalistic decisions have some similarity to parametric working memory tasks [19,20,36,37] where subjects compare two different levels of input via the same afferent pathway.

Our model is state-based, behaving similarly to models of bistable percepts [6,38] in that the neural activity can transition at a stimulus-related but non-deterministic time from one state to another, even if the stimulus does not change. Evidence for such noise-induced transitions between otherwise stable states has been obtained across a number of neural systems [5,39–44], even where trial-averaged activity may suggest a slower ramping akin to evidence accumulation [10]. Indeed, a similar state transition may underlie the more commonly studied perceptual decision-making tasks [9].

While our model does not designate the location of the simulated circuit, we assume all neurons in our model are located in the same region and are likely to be found in anterior cingulate cortex, where neurons represent value in self-paced decisions [45], or a subregion of the ventromedial prefrontal cortex, such as the infralimbic cortex, whose activity is needed for an animal to engage in feeding behavior [46].

## The nature of competition

Decision making inevitably involves competition in that choices have opportunity cost. The more often one alternative is chosen, the less often another alternative can be chosen. In winner-take-all networks the competition is inherent in the network's structure and a binary decision arises on a trial-by-trial basis [14,25]. That is, at the level of neural activity, increased spiking of one group of cells inhibits the spiking of other groups [11,14]. At the level of behavior one choice terminates the trial and prevents any other choice.

By contrast, in a food preference test implicit competition can arise in the absence of any correlation between the neural activity in response to one stimulus versus the other. This is simply due to the finite amount of time available, such that if one stimulus is extremely hedonic the animal will rarely leave it, so inevitably consume less of any alternative stimulus. That is, even if the bout duration of an alternative stimulus is unchanged, an animal would leave a more hedonic stimulus less frequently to visit that alternative, so consume less of the alternative than if that alternative were paired with a less hedonic stimulus.

Theoretically, it is possible for a highly hedonic stimulus to boost the hedonic value of an alternative stimulus while still maintaining such implicit competition between the total amounts of the two foods being consumed. Such behavior could arise if, for example, the neural activity were dominated by a slow synaptic facilitation such that the impact of a highly

hedonic response lingered so as to boost responses to later stimuli. However, to align with the qualitative nature of the marginal value theorem [2,47]—one location is more easily left if the alternatives are better—we incorporated synaptic depression as the dominant effect of the network on a timescale of many seconds.

Synaptic depression—an inevitable consequence of synaptic transmission as vesicles are depleted—causes a reduction in the hedonic response to any stimulus that follows a highly hedonic stimulus. In this manner, synaptic depression accentuates competition between two taste stimuli in a food preference test, because the mean bout duration when sampling the less hedonic stimulus is shortened in the presence of a highly hedonic alternative stimulus. Such competition based on a trace-memory of prior stimuli is in addition to the implicit competition already present in a food preference task. We find that the reduction in bout durations in the presence of a highly hedonic alternative is most pronounced, with a ten-fold reduction possible, in the "entice to stay" networks we simulated here.

Our goal here is to assess what behavioral phenomena arise from a very simple decision-making circuit when common neuronal features are incorporated. The short-term synaptic depression we include is one of many biological processes that sculpt network activity over multiple timescales. Short-term depression allows our network to be reactive to the recent past, so could account for anhedonia in a short period following a strongly rewarding stimulus [48]. However, it is insufficient to produce the longer-term learning needed for an animal to anticipate the future based on past experience, as in the anticipatory contrast effect [49]. It is intriguing however, that the basic phenomenon of the devaluing of a rewarding stimulus (such as saccharine) in circumstances when a more highly rewarding or addictive stimulus is available or anticipated (such as cocaine or sucrose)[50–52], arises in our relatively simple circuit.

Our networks reproduce many effects underlying the marginal value theorem [2,47], which provides a framework for behavior observed in the foraging literature [35]. According to the marginal value theorem, an animal stays at a patch of food until its rate of return diminishes to the average rate of return obtained by moving from site to site across the environment. While we do not simulate an experiment with reduced stimulus over time during a bout, our model does reproduce some of the qualitative features essential in any neural circuit model of such foraging dynamics. First, the more hedonic/palatable the current stimulus, the greater the bout duration at that stimulus. Second, the more hedonic/palatable the alternative stimulus, the shorter the bout duration at the first stimulus. Third, the greater the time in the switching period between stimuli, the less the impact of a more hedonic alternative. Importantly, the third effect mitigates the second effect; longer times spent between stimuli will mitigate shorter sampling bouts in the presence of superior alternatives. Such reluctance to switch to more favorable alternatives arises from the cost of switching in value-based models, because animals do not accumulate reward while traveling between samples. Here we show that such behaviors, desirable from the point of view of optimality, arise qualitatively in a simple circuit with depressing synapses.

## Methods

### Properties of model neurons

Individual neurons were simulated with an exponential leaky integrate-and-fire model [53] following the equation:

$$C_m \frac{dV_m}{dt} = \frac{E_l - V_m + \Delta_{th}\exp\left(\frac{V_m - V_{th}}{\Delta_{th}}\right)}{R_m} + G_{syn}S_I\left(E_{rev_I} - V_m\right) + G_{syn}S_E\left(E_{rev_E} - V_m\right)$$
$$+ G_{ref}(E_K - V_m) + G_{ext_I}\left(E_{rev_I} - V_m\right) + G_{ext_E}\left(E_{rev_E} - V_m\right)$$

where $V_m$ is the membrane potential, $C_m$ is the total membrane capacitance, $E_l$ is the leak potential, $R_m$ is the total membrane resistance, $\Delta_{th}$ is the spiking range, $V_{th}$ is the spiking threshold, $S$ is the synaptic input variable, $G_{syn}$ and $E_{rev}$ are the maximal conductance and reversal potential for synaptic connections, $G_{ref}$ is the dynamic refractory conductance, $E_K$ is the potassium reversal potential, and $G_{ext}$ is the input conductance. The "E" and "I" subscripts denote the variables specific to excitatory and inhibitory channels, respectively (e.g. $S_E$ and $E_{rev_E}$ are the synaptic input and reversal variables for excitatory channels; $S_I$ and $E_{rev_I}$ are the corresponding inhibitory variables). This equation simulates the neuron's membrane potential until $V_m > V_{spike}$, at which point the neuron spikes.

When a neuron spikes, $V_m$ is set to the $V_{reset}$ value. Additionally, the neuron's refractory conductance, synaptic output, $s$, and synaptic depression (noted as $D$) are updated according to the equations:

$$G_{ref} \mapsto G_{ref} + \Delta G_{ref}$$

$$s \mapsto s + p_R D_{fast}(1 - s)$$

$$D_{fast} \mapsto D_{fast}(1 - p_R)$$

where $\Delta G_{ref}$ is the increase in refractory conductance, and $p_R$ is the vesicle release probability following a spike.

In the timestep immediately following a spike, the neuron's membrane potential continues to follow the exponential leaky integrate-and-fire model equation. In this equation the separate excitatory ($S_{E,i}$) and inhibitory ($S_{I,i}$) synaptic inputs for cell $i$ are obtained from the sum of all presynaptic outputs multiplied by the corresponding connection strengths, $W_{ij}$, from neurons $j$ (see *Network architecture and connections*):

$$S_i = \sum_j W_{ij} s_j,$$

each of which decay with the appropriate (excitatory or inhibitory) synaptic gating time constant $\tau_S$ according to:

$$\frac{ds_i}{dt} = -\frac{s_i}{\tau_S}.$$

Likewise, refractory conductance decays with the time constant $\tau_{ref}$ according to:

$$\frac{dG_{ref}}{dt} = -\frac{G_{ref}}{\tau_{ref}}$$

The $G_{ext}$ input conductance serves as both noisy-background and stimulus inputs in the same manner. Inputs were modeled as Poisson spike trains with rates $r_{noise}$ and $r_{stimulus}$, which produce input spikes (from all sources) at timepoints $\{t_{sp}\}$. Please note, the noisy-background includes both excitatory and inhibitory spiking input (included in $G_{ext_I}$ and $G_{ext_E}$, respectively); the $r_{noise}$ parameter specifies the rate for both excitatory and inhibitory background noise. The input conductance values for a given timepoint, $t$, are updated as:

$$G_{ext} \mapsto G_{ext} + \Delta G_{ext} \delta(t - t_{sp})$$

where the conductance increases by $\Delta G_{ext}$ at the time of each input spike. The input

    

conductance otherwise decays with the time constant $\tau_{ext}$ according to:

$$\frac{dG_{ext}}{dt} = -\frac{G_{ext}}{\tau_{ext}}.$$

The cellular parameters with values specific to excitatory neurons (e.g. that differ from inhibitory values) are: $E_{rev_E} = 0\ mV$, $\tau_s = 50\ ms$, and $\tau_{ext} = 3.5\ ms$. The complementary values for inhibitory neurons are: $E_{rev_I} = -70\ mV$, $\tau_s = 10\ ms$, and $\tau_{ext} = 2\ ms$. The remaining parameters applicable to both excitatory and inhibitory neurons are: $G_{syn} = 10\ nS$, $p_R = .1$, $\tau_{fast} = 300\ ms$, $\tau_{slow} = 7\ s$, $p_{slow} = .5$, $E_l = -70\ mV$, $E_K = -80\ mV$, $V_{reset} = -80\ mV$, $R_m = 100\ M\Omega$, $C_m = 100\ pF$, $V_{spike} = 20\ mV$, $\Delta G_{ext} = 1\ nS$, $V_{th} = -50\ mV$, $\Delta_{th} = 2\ mV$, $\tau_{ref} = 25\ ms$, and $\Delta G_{ref} = 12.5\ nS$. The Poisson spike-train parameters $r_{noise}$ and $r_{stimulus}$ are described in the next section. Neurons were simulated with a simulation timestep $dt = .1\ ms$.

## Synaptic depression

We modeled synaptic depression using two separate timescales, noted in the previous spike-update equations as $D_{slow}$ and $D_{fast}$. These two variables reflect, respectively, the fraction of the maximum number of vesicles available in the reserve pool and the release-ready pool. Following a spike, the variables recover to a value of one with different timescales, because vesicles regenerate and are replenished slowly in the reserve pool, but may dock and become release-ready much more quickly once available in the reserve pool (Fig 7).

Specifically, $D_{slow}$ represents the ratio of currently available reserve-pool vesicles out of the maximum possible, that is $D_{slow} = \frac{N_{pool}}{N_{max}}$ (Fig 7A). These dock quickly at empty docking sites on the timescale $\tau_{fast}$ (Fig 7C), but are replaced slowly on the timescale $\tau_{slow}$. $D_{fast}$ represents the ratio of docked vesicles out of total docking sites, that is $D_{fast} = \frac{N_{docked}}{N_{sites}}$ (Fig 7B). We also incorporate the constant parameter, $f_D = 0.05$, which is equal to the ratio of the number of docking sites to the maximum size of the reserve pool of vesicles, $f_D = \frac{N_{sites}}{N_{max}}$. Only docked vesicles can be released immediately following a spike, such that upon each spike we update $D_{fast} \mapsto D_{fast}(1 - p_R)$ where $p_R$ is the vesicle release probability.

During sustained spiking, the fast-docking can maintain a firing-rate dependent supply of docked vesicles until the reserve pool (Fig 7B) depletes. Vesicles dock at empty sites (Fig 7C) according to:

$$\frac{dD_{fast}}{dt} = \frac{(D_{slow} - D_{fast})}{\tau_{fast}}$$

Reserve-pool vesicles fill the empty docking sites on the fast timescale $\tau_{fast}$. On the other hand, the reserve-pool regenerates much more slowly according to:

$$\frac{dD_{slow}}{dt} = \frac{(1 - D_{slow})}{\tau_{slow}} - f_D \frac{(D_{slow} - D_{fast})}{\tau_{fast}}$$

The first term represents the reserve-pool vesicle regeneration on timescale $\tau_{slow}$. The second term $-f_D \frac{(D_{slow} - D_{fast})}{\tau_{fast}}$ accounts for the vesicles lost due to docking (Fig 7C).

Our model reflects the empirical evidence showing the effects of synaptic-depression at short timescales on the order of milliseconds, and longer timescales on the order of seconds [54,55]; depression timescales on the order of minutes have even reported in non-mammalian animals [56]. Additional, recent evidence [57] directly supports our fast-depression

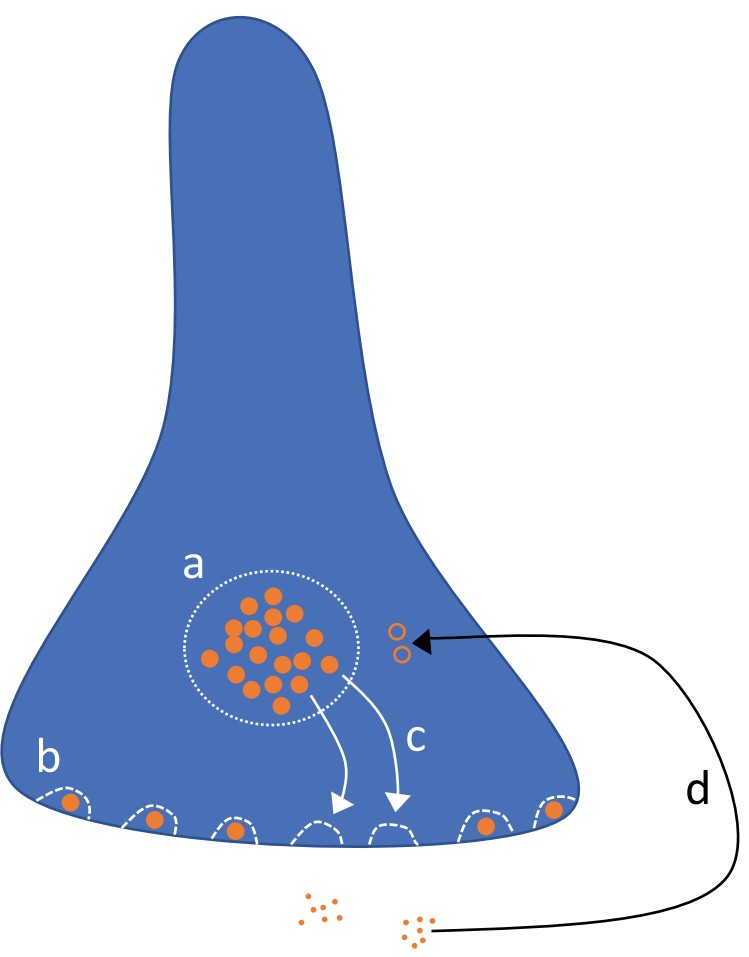

**Fig 7. Synaptic depression model.** Vesicles are either in the reserve pool (a) or at docking sites (b). Available vesicles dock quickly at empty docking sites (c), but take much longer to regenerate once their internal neurotransmitter is released (d).

mechanism where available vesicles quickly refill empty docking sites. Our model provides a coherent mechanism for both fast-acting and long-lasting synaptic depression effects.

## Network architecture and connections

Each network consists of 250 individual neurons, split into two populations of 100 excitatory cells (*i. e.*, "stay" and "switch" populations, $E_{stay}$ and $E_{switch}$) and two populations of 25 inhibitory cells ($I_{stay}$ and $I_{switch}$). Fig 1A depicts the basic architecture of all networks simulated in this paper, with connections within and between populations indicated. For each pair of connected populations (or for self-connected excitatory populations) pairs of cells were connected probabilistically with a probability, $P(connection)$ = .5. The strength of connections were symmetric across "stay" and "switch" populations, but depended on whether presynaptic or postsynaptic cells were excitatory or inhibitory, as indicated in Table 1.

## Code availability

The code used to simulate our model is freely available online at https://github.com/johnksander/naturalistic-decision-making

**Table 1. Model neuron parameters.**

| Name | Description | value |
|---|---|---|
| $E_{rev}$ | Reversal potential | Excitatory cells: 0 mV Inhibitory cells: −70 mV |
| $E_l$ | Leak potential | −70 mV |
| $E_K$ | Potassium potential | −80 mV |
| $R_m$ | Membrane resistance | 100 MΩ |
| $C_m$ | Membrane capacity | 100 pF |
| $\tau_s$ | Synaptic gating timescale | Excitatory cells: 50 ms Inhibitory cells: 10 ms |
| $V_{reset}$ | Reset membrane potential | −80 mV |
| $V_{spike}$ | Spike threshold | 20 mV |
| $\tau_{ext}$ | noisy-background conductance timescale | Excitatory cells: 3.5 ms Inhibitory cells: 2 ms |
| $G_{syn}$ | Synaptic max conductance | 10 nS |
| $\tau_{fast}$ | Fast depression timescale (Fig 7C) | 300 ms |
| $\tau_{slow}$ | Slow depression timescale (Fig 7D) | 7 s |
| $P_R$ | Vesicle release probability | .1 |
| $f_D$ | Ratio of max docked vesicles to max pooled vesicles | .05 |
| $D_{fast}$ | Ratio of docked vesicles out of total possible (Fig 7B) | $\frac{N_{docked}}{N_{sites}}$ |
| $D_{slow}$ | Ratio of reserve-pool vesicles out of the total possible (Fig 7A) | $\frac{N_{pool}}{N_{max}}$ |
| $\Delta G_{ext}$ | Conductance step-increase to external input spike | 1 nS |
| $V_{th}$ | exponential spiking-term threshold | −50 mV |
| $\Delta_{th}$ | spiking range | 2 mV |
| $\tau_{ref}$ | Refractory conductance timescale | 25 ms |
| $\Delta G_{ref}$ | Step change in refractory conductance | 12.5 nS |
| $dt$ | Simulation timestep | .1 ms |

**A Model summary**

| | |
|---|---|
| Populations | Stay: 1 excitatory, 1 inhibitory<br>Leave: 1 excitatory, 1 inhibitory |
| Connectivity | Within-pool (stay or leave): I-to-E and recurrent E-to-E<br>Cross-pool (stay-to-leave or leave-to-stay): E-to-I |
| Neuron model | Exponential Leaky Integrate and Fire (ELIF) with dynamic refractory conductance |
| Synapse model | Conductance based, step increase followed by exponential decay |
| Plasticity | Depression with two timescales |
| Input | Noisy background input: fixed-rate Poisson spike trains to all cells<br>Stimuli: Poisson spike trains to E-stay and E-leave cells |
| Measurements | Spike trains, activity state-durations, connection strengths |

**B Populations**

| Name | Elements | Size |
|---|---|---|
| E-stay | ELIF neurons | 100 |
| I-stay | ELIF neurons | 25 |
| E-leave | ELIF neurons | 100 |
| I-leave | ELIF neurons | 25 |
| Noisy background input | Poisson trains | 500 |
| Aversive stimulus | Poisson trains | 100 |
| Hedonic stimulus | Poisson trains | 100 |

*(Continued)*

## C Connectivity

| Name | Source | Target | Pattern | |
|---|---|---|---|---|
| E-to-I | E-stay<br>E-leave | I-leave<br>I-stay | Random, $p = .5$, model-dependent fixed weight: | |
| | | | Circle (open) | 0.2955 |
| | | | Circle (closed) | 0.0833 |
| | | | Square (open) | 0.4242 |
| | | | Square (closed) | 0.0909 |
| | | | Up-triangle (open) | 0.75 |
| | | | Up-triangle (closed) | 0.75 |
| | | | Diamond (open) | 0.4773 |
| | | | Diamond (closed) | 0.4621 |
| | | | Down-triangle (open) | 0.4697 |
| | | | Down-triangle (closed) | 0.1742 |
| I-to-E | I-stay<br>I-leave | E-stay<br>E-leave | Random, $p = .5$, model-dependent weight: | |
| | | | Circle (open) | 12.3747 |
| | | | Circle (closed) | 12.3747 |
| | | | Square (open) | 9.4939 |
| | | | Square (closed) | 9.6192 |
| | | | Up-triangle (open) | 8.4919 |
| | | | Up-triangle (closed) | 3.6071 |
| | | | Diamond (open) | 9.4939 |
| | | | Diamond (closed) | 3.6071 |
| | | | Down-triangle (open) | 8.8677 |
| | | | Down-triangle (closed) | 4.2333 |
| E-to-E | E-stay, E-leave | E-stay, E-leave | Random, $p = .5$, fixed weight, $W^{EE} = 0.0405$ | |

## D Neuron and Synapse Model

| | |
|---|---|
| **Name** | Laf neuron |
| **Type** | Dynamic leaky integrate-and-fire with dynamic refractory conductance |

**Subthreshold dynamics**

$$C_m \frac{dV_m}{dt} = \frac{E_L - V_m + \Delta_{th}\exp\left(\frac{V_m - V_{th}}{\Delta_{th}}\right)}{R_m} + G_{syn} \cdot S_I \left(E_{rev_I} - V_m\right) + G_{syn} \cdot S_I \left(E_{rev_I} - V_m\right) + G_{syn} \cdot S_E \left(E_{rev_E} - V_m\right) + G_{ref}(E_K - V_m) + G_{ext_I}\left(E_{rev_I} - V_m\right) + G_{ext_E}\left(E_{rev_E} - V_m\right)$$

$$\frac{dG_{ref}}{dt} = -\frac{G_{ref}}{\tau_{ref}}$$

$$\frac{dG_{ext}}{dt} = -\frac{G_{ext}}{\tau_{ext}}$$

**Spiking**

If $V_m > V_{spike}$:
1. Emit spike with timestamp $t$
2. $G_{ref} \mapsto G_{ref} + \Delta G_{ref}$
3. $V_m \mapsto V_{reset}$

**Synapse**

$$S_i = \sum_j W_{ij} \hat{s}_j$$

following a spike by neuron $i$:

$$s_I \mapsto s_I + p_R D_{fast}(1-s_I)$$

$$D_{fast,I} \mapsto D_{fast,I}(1-p_R)$$

Between spikes:

$$\frac{ds_I}{dt} = -\frac{s_I}{\tau_S}$$

$$\frac{dD_{fast,I}}{dt} = \frac{(D_{slow,I} - D_{fast,I})}{\tau_{fast}}$$

$$\frac{dD_{slow,I}}{dt} = \frac{(1 - D_{slow,I})}{\tau_{slow}} - f_D \frac{(D_{slow,I} - D_{fast,I})}{\tau_{fast}}$$

**F Input**

| Type | Description |
| --- | --- |
| All external spiking input | Input spikes increase conductance: $G_{ext} \mapsto G_{ext} + B \cdot \Delta G_{ext}$<br>Conductance $G_{ext}$ decays:<br>$\frac{dG_{ext}}{dt} = -\frac{G_{ext}}{\tau_{ext}}$ |
| Background noisy input | One excitatory spike-train per neuron, and one inhibitory spike-train per neuron (all 1540 Hz Poisson spike-trains). |
| Aversive stimulus | One excitatory spike-train per E-leave neuron (fixed rate). |
| Hedonic stimulus | One excitatory spike-train per E-stay neuron (fixed rate). |

**G Measurements**

**Active state: when mean difference between E-stay and E-leave excitatory synaptic gating exceeds .02 for 50ms (consecutively).**

**State duration/sampling duration: time between state transitions (i.e. transitioning from E-stay to E-leave active state).**

## Network states and stimuli

A network's active state was evaluated by comparing the mean values of synaptic output, $s_E$, averaged across all excitatory cells in each of the two excitatory populations. Specifically, when the difference between the mean output of the previously less active excitatory population exceeded that of the previously more active excitatory population by a threshold of 0.02 consistently for 50ms, we recorded a state change. The threshold was chosen to be .02 as a value which robustly captured the intended quasi-bistable behavior across multiple networks. If a simulation produced one second of consecutive timepoints without an active pool (*i. e.*, because the difference in activity of the two excitatory populations was too small to produce above-threshold differences in synaptic output) the simulation was terminated and the corresponding parameter set was not used for further analysis. We only analyzed simulations of networks exhibiting quasi-bistability

In our simulations of preference tests, we did not simulate the animal's behavior in between bouts of sampling a stimulus. Once the excitatory neurons in the "switch" population (E-switch cells) were recorded as more active than those in the "stay" population, using the threshold mentioned above, we removed the stimulus input to the network. 100 ms later, we induced a subsequent transition back to the "stay" state to represent the animal initiating a new bout of stimulus sampling. The transition back to sampling was accomplished by halving the noisy background input to E-switch cells until the network transitioned again to the "stay" state. At all other times in simulations, the noisy background input remained constant. Once a transition to the "stay" state was recorded (by excitatory cells in the "stay" population being more active than those in the "switch" population) input stimulus was applied to indicate the next bout of sampling.

## Parameter sweep and example networks

We obtained a set of examplar networks by varying I-to-E and E-to-I connection strengths and recording the average active-state durations. Networks were simulated without stimulus (but with noisy-background input) for 1500 seconds. We then chose five pairings of intrinsically fast-transitioning and intrinsically slow-transitioning networks from this parameter space. The fast and slow networks represent the "entice-to-stay" and "repel-to-leave" decision-making accounts, respectively. Networks were paired across the two classes by selecting examples with connection strengths as closely matched as possible, while average state-durations were either end of the wide range recorded. Typically, fast networks transitioned in a few seconds, whereas slow networks transitioned in a few minutes. We performed all subsequent simulations with 10 exemplar networks (five pairs) as represented by the symbols in Fig 2.

For each of the 10 exemplar networks, we determined the level of stimulus necessary to equate the baseline sampling behavior, producing an average duration of 7.5 seconds for the "stay" state in each case. Increasing the strength of hedonic stimuli (increasing E-stay spiking input) slowed the fast network transitions, while increasing the strength of aversive stimuli (increasing E-leave spiking input) quickened slow network transitions. The specific intensities were found with Matlab's fminbnd() optimizer function. We will refer to these stimulus values as the networks' "baseline stimuli", as we used these values as reference points for subsequent simulations.

## Taste preference task simulations

Individual taste preference task simulations lasted 1500 seconds total. Each simulation compared sampling bout durations in response to two stimuli (A and B) each with a fixed value across the session. The input representing stimulus A was equal to the network's baseline

stimulus value, and constant across sessions. The input representing stimulus B was systematically varied across sessions, such that in each session it was fixed at a value between zero and sixty times the input for stimulus A. For a given network the stimulus inputs targeted the same population for all sessions. The stimulus value (A or B) alternated after each stay-to-leave transition. Simulations always began with stimulus A during the first stay-state, followed by stimulus B during the second stay-state, etc. This represents the animal alternating between the available stimuli after "leave" decisions.

## Supporting information

**S1 Fig. Firing rates of active cells in the "stay" state as a function of network connection strengths.** (Left) Firing rate of excitatory cells is in the vicinity of 10 Hz for all parameters leading to two quasi-stable network states. (Right) Firing rate of inhibitory cells varies strongly as a function of parameters, but note that the range of rates is similar for entice-to-stay networks (left edge) and repel-to-leave networks (right edge) so firing rate is not a clear indicator of type of network.
(TIFF)

**S2 Fig. State durations in an alternative network to indicate robustness of the system to incorporation of additional connections.** The results are produced for a circuit depicted in A) with additional connections (excitatory between excitatory pools and inhibitory both within and between inhibitory pools) all at half the strength of the excitatory within-pool connections of the standard network. B) Results indicate the same network responses as in the original network are possible given small compensatory shifts in the inhibitory-to-excitatory and excitatory-to-inhibitory connections (compare main manuscript, Fig 2).
(EPS)

**S3 Fig. Competitive interaction in taste preference tests.** All network results are shown, with marker symbols identifying the network parameters as depicted in Fig 2 (main text Fig). Results for all of the intrinsically fast-switching (entice-to-stay) networks exhibited a stronger impact on the duration of bouts at the stimulus of fixed input, A, as the strength of input from the alternative stimulus, B, was adjusted to reflect more hedonic input.
(TIFF)

**S4 Fig. Impact of varying one stimulus input on sampling durations.** In all panels, input corresponding to stimulus B is adjusted across sessions, while input corresponding to stimulus A is held fixed. Mean bout durations in response to stimulus B are shown as dashed lines and increase with increasing input in entice-to-stay networks (left panels), while they decrease with increasing input in repel-to-leave networks (right panels). Mean bout durations in response to stimulus A (which never changes across sessions) are shown as solid lines. Competition arises as durations for A (solid lines) decrease when durations for B (dotted lines) increase, with a much stronger effect in all of the entice-to-stay networks (left panels).
(TIFF)

## Author Contributions

**Conceptualization:** Donald B. Katz, Paul Miller.

**Data curation:** John Ksander.

**Formal analysis:** John Ksander.

**Funding acquisition:** Donald B. Katz, Paul Miller.

**Methodology:** Paul Miller.

**Software:** John Ksander.

**Supervision:** Donald B. Katz, Paul Miller.

**Visualization:** John Ksander.

**Writing – original draft:** John Ksander.

**Writing – review & editing:** Donald B. Katz, Paul Miller.

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
