## [Decision Letter · Decision Letter 0]

5 Jun 2021

Dear Dr. Miller,

Thank you very much for submitting your manuscript "A model of naturalistic decision making in preference tests" for consideration at PLOS Computational Biology.

As with all papers reviewed by the journal, your manuscript was reviewed by members of the editorial board and by several independent reviewers. In light of the reviews (below this email), we would like to invite the resubmission of a significantly-revised version that takes into account the reviewers' comments.

We cannot make any decision about publication until we have seen the revised manuscript and your response to the reviewers' comments. Your revised manuscript is also likely to be sent to reviewers for further evaluation.

Sincerely,

Alireza Soltani

Associate Editor

PLOS Computational Biology

Samuel Gershman

Deputy Editor

PLOS Computational Biology

Reviewer's Responses to Questions

**Comments to the Authors:**

Reviewer #1: The authors present a very interesting and clear set of simulations exploring switching and decision making networks. The methods are clear, results coherent and well introduced. What is lacking is a discussion as to how this applies to brain areas in general or specifically. The networks are simple feedback excitatory networks with global inhibition allowing to exhibit attractor states and to have these be suppressed. While each individual module could easily represent a cortical area or part of one, I don't know of any cortical areas that interact with each other by exciting the other networks inhibitory neurons (with the exception of olfactory cortex projections to olfactory bulb). I would encourage the authors to add some discussion on where and how these phenomena would be implemented either as a general or taste preference specific mechanism. As it is one is left with a rather simplistic analysis of mutual interactions between attractor networks. Which are very interesting in the context presented but need integration into the biological question.

Reviewer #2: The present work addressed decision making processes in the interesting context of a preference task. The computational models are proposed to underlie two opposing strategies that animals might follow in taste preference tests: one in which the animal prefers to switch unless the current stimulus is highly appealing ('enticed to stay') and another in which the animal prefers to stay with its current choice, unless it is repelled from such choice by the attractiveness of other available options ('repel to leave'). The study of these models leads to behavioral and electrophysiological predictions to discern between both strategies.

I think that the study is quite original and tackles a highly interesting topic. I have however one concern that diminished my enthusiasm about the presented results (plus a few others that should also be addressed):

1) While the paper is overall well written, and the models themselves are standard spiking networks, the interpretation given to such models and their components is extremely hard to follow and at points inconsistent. This makes the paper really unclear and potentially misleading in its conclusions. Inconsistencies appear across several figures and results:

-In figure 1, we are presented with a first version of the model, in which one excitatory population encodes the option 'stay' (blue in the figure) and the other one encodes the option 'switch' (orange). Panel B shows that both populations can sustain their activity for several seconds --however, I have a hard time understanding 'switch' as a choice that is sustained for long periods, rather than a sharp signal triggering a motor action. What's the meaning of a populations which persistently encodes 'switch' over several seconds?

-Figure 3 then shows that the duration of the 'switch' option is extremely short, contradicting what the authors showed in Fig. 1. This is due, the authors explain later, to the fact that the simulation is reset after each switch. This makes sense as it prevents the 'switch' population from displaying sustained activity for long periods, although it makes the model discontinuous from one choice to the next.

-The work then focuses on the effects that the duration of a given choice ('stimulus A') has on the duration on the next chosen option (say, stimulus B). This is precisely one of the only things that the model, as explained up to Fig. 3, should be unable to do given the hard reset after a decision to switch. The authors reinterpret the model again to accommodate this continuous simulation of 1500 seconds, by effectively swapping the populations 'stay' and 'switch' after each switch. This is an extremely complicated way of simulating something much simpler: a competition between stimulus A and B which cross-inhibit each other (something that, on the other hand, is not as novel as the proposed topic in which behavioral conditions 'stay' and 'switch' are the opposing options rather than two input choices).

If the authors would like to model the competition between staying and switching for a taste preference task, a much cleaner way would be to build a model with two well-defined excitatory populations (encoding taste 'A' and taste 'B') and a third population with sharp, transient responses for the 'switch' command. In the present framework, on the other hand, it is very unclear what each of the populations encodes along all the simulation, and arbitrarily swapping labels may invalidate results about inter-trial memory effects sought by the authors.

Other comments:

2) While the authors did a good job taking into account the effects of variability in E-to-I and I-to-E synapses, it would be interesting to see whether the model is robust to physiologically plausible connections not present in the model, such as E-to-E (both recurrent and reciprocal) and I-to-I.

3) Fig 1: panel D and C should be exchanged, as D is introduced and explained earlier in the text than C.

4) Fig 2: aren't panels A and B the same? If so, authors could drop panel A, as the symbols in B are not really occluding much of the figure.

5) Page 10: This sentence needs a citation: "durations between eight and nine seconds. These are typical bout durations for a rat licking water from a spout".

6) Page 17, lines 263-5: "intrinsically slow switching (entice-to-stay) networks (left) and intrinsically fast-switching (repel-to-leave) networks (right)". Should the fast switching network be the entice-to-stay, and the slow one the repel-to-leave?

**Have the authors made all data and (if applicable) computational code underlying the findings in their manuscript fully available?**

Reviewer #1: Yes

Reviewer #2: **No: **The code has not been available and there is no indication in the manuscript that it will be made available upon publication.

PLOS authors have the option to publish the peer review history of their article (what does this mean?). If published, this will include your full peer review and any attached files.

Reviewer #1: No

Reviewer #2: No
---

## [Decision Letter · Decision Letter 1]

10 Sep 2021

Dear Dr. Miller,

We are pleased to inform you that your manuscript 'A model of naturalistic decision making in preference tests' has been provisionally accepted for publication in PLOS Computational Biology.

Best regards,

Alireza Soltani

Associate Editor

PLOS Computational Biology

Samuel Gershman

Deputy Editor

PLOS Computational Biology

Reviewer's Responses to Questions

**Comments to the Authors:**

Reviewer #1: authors have addressed my comments

**Have the authors made all data and (if applicable) computational code underlying the findings in their manuscript fully available?**

Reviewer #1: Yes

PLOS authors have the option to publish the peer review history of their article (what does this mean?). If published, this will include your full peer review and any attached files.

Reviewer #1: **Yes: **CHRISTIANE LINSTER

---

## [Editor Report · Acceptance letter]

16 Sep 2021

PCOMPBIOL-D-21-00701R1 

A model of naturalistic decision making in preference tests

Dear Dr Miller,

I am pleased to inform you that your manuscript has been formally accepted for publication in PLOS Computational Biology. Your manuscript is now with our production department and you will be notified of the publication date in due course.

With kind regards,

Andrea Szabo
